# Beyond HOMA-IR: Comparative Evaluation of Insulin Resistance and Anthropometric Indices Across Prediabetes and Type 2 Diabetes Mellitus in Metabolic Syndrome Patients

**DOI:** 10.3390/life15121845

**Published:** 2025-11-30

**Authors:** Mohamed-Zakaria Assani, Lidia Boldeanu, Anda Lorena Dijmărescu, Daniel Cosmin Caragea, Ionela Mihaela Vladu, Diana Clenciu, Adina Mitrea, Alexandra-Ștefania Stroe-Ionescu, Mariana-Emilia Caragea, Isabela Siloși, Mihail Virgil Boldeanu

**Affiliations:** 1Doctoral School, University of Medicine and Pharmacy of Craiova, 200349 Craiova, Romania; mohamed.assani@umfcv.ro (M.-Z.A.); alexandra.stroe@yahoo.com (A.-Ș.S.-I.); mariana.emilia77@yahoo.com (M.-E.C.); 2Department of Immunology, Faculty of Medicine, University of Medicine and Pharmacy of Craiova, 200349 Craiova, Romania; isabela_silosi@yahoo.com (I.S.); mihail.boldeanu@umfcv.ro (M.V.B.); 3Department of Microbiology, Faculty of Medicine, University of Medicine and Pharmacy of Craiova, 200349 Craiova, Romania; lidia.boldeanu@umfcv.ro; 4Department of Obstetrics and Gynecology, Faculty of Medicine, University of Medicine and Pharmacy of Craiova, 200349 Craiova, Romania; 5Department of Nephrology, Faculty of Medicine, University of Medicine and Pharmacy of Craiova, 200349 Craiova, Romania; 6Department of Diabetes, Nutrition and Metabolic Diseases, Faculty of Medicine, University of Medicine and Pharmacy of Craiova, 200349 Craiova, Romania; ionela.vladu@umfcv.ro (I.M.V.); dianaclenciu@yahoo.com (D.C.); ada_mitrea@yahoo.com (A.M.)

**Keywords:** insulin resistance, metabolic syndrome, HOMA-IR, METS-IR, TyG, PreDM, T2DM, anthropometric indices

## Abstract

Insulin resistance is central in metabolic syndrome, but indices such as Homeostasis Model Assessment-estimated Insulin Resistance (HOMA-IR) require insulin assays that are costly and not always available. Non-insulin-based indices and refined anthropometric markers may offer simpler risk stratification in prediabetes and diabetes. Our objective was to compare insulin and non-insulin-based indices of insulin resistance, together with advanced anthropometric and lipid markers, between prediabetes (PreDM) and type 2 diabetes (T2DM) and across hypertension grades in metabolic syndrome. We conducted a cross-sectional study in 200 adults with metabolic syndrome, 80 with PreDM and 120 with T2DM. Clinical, anthropometric and biochemical parameters were recorded, and HOMA-IR, Homeostasis Model Assessment of Beta-cell function (HOMA%B), Metabolic Score for Insulin Resistance (METS-IR), triglyceride to glucose index (TyG), triglyceride-to-glucose index to high-density lipoprotein cholesterol ratio (TyG/HDL-c) and other derived indices were calculated. Group comparisons, correlations and multiple linear regression were performed. Compared with PreDM, T2DM showed higher glycemic indices and inflammation, but similar body mass index (BMI) and triglycerides. Across glycemic categories and hypertension grades, METS-IR, TyG and TyG/HDL-c increased and correlated strongly with body roundness index (BRI), abdominal volume index (AVI) and weight-adjusted waist index (WWI), while HOMA-IR contributed little independent information. In regression models, lipid adipose product (LAP) and WWI best explained METS-IR in prediabetes, whereas TyG and BRI were the main determinants of METS-IR in diabetes. In metabolic syndrome with PreDM or T2DM, METS-IR and TyG, particularly combined with BRI, AVI and WWI, outperformed traditional lipid ratios and added value beyond HOMA-IR. These composite indices appear useful for insulin resistance assessment when insulin measurement is unavailable or unreliable.

## 1. Introduction

Insulin resistance (IR) is increasingly recognized as a fundamental pathophysiological nexus underlying type 2 diabetes mellitus (T2DM), cardiovascular disease (CVD), non-alcoholic fatty liver disease (NAFLD), and metabolic syndrome (MetS). Conventional indices such as homeostatic model assessment of insulin resistance (HOMA-IR) and triglyceride-to-glucose index (TyG) rely on insulin or fasting glucose concentrations, which pose challenges for routine clinical implementation owing to variability, cost, and patient burden. This has catalyzed the development of novel, non-insulin-based surrogate markers leveraging routinely available laboratory and anthropometric data, including metabolic score for insulin resistance (METS-IR), Body Roundness Index (BRI), Waist-to-Height Ratio (WHtR), and Weight-Adjusted Waist Index (WWI) [1,2,3,4,5,6].

Diabetes has emerged as one of the most rapidly increasing global health challenges of the 21st century. According to the 10th edition of the International Diabetes Federation (IDF) Diabetes Atlas, the estimated prevalence was approximately 537 million individuals in 2021. Projections indicate a rise to 643 million by 2030 and potentially reaching 783 million by 2045. Prediabetes (PreDM) serves as a critical antecedent to T2DM, characterized by glycemic levels that exceed normoglycemia yet fall below diagnostic thresholds for diabetes. It represents an intermediate hyperglycemic state significantly associated with increased risk of T2DM onset. The progression from prediabetes to overt diabetes is typically insidious, occurring over several years. Notably, once established, diabetes is often viewed as a chronic, largely irreversible condition [7,8,9,10,11].

MetS represents a significant global public health challenge, encompassing over one billion adults across both industrialized and emerging economies. It is characterized by a constellation of interrelated metabolic derangements, including central adiposity, insulin resistance (IR), hypertension, and dyslipidemia. This clustering markedly elevates the risk for CVD, T2DM, and all-cause mortality. Patients with MetS exhibit a substantially increased risk of acute cardiovascular events such as myocardial infarction and cerebrovascular incidents. Central to MetS pathophysiology, IR is defined by a diminished cellular sensitivity to insulin, resulting in decreased glucose uptake and hyperglycemia. Beyond its role in T2DM pathogenesis, IR serves as an independent prognostic indicator for adverse cardiovascular outcomes, even in individuals without explicit diabetic diagnoses. [12,13,14,15,16,17,18,19,20,21].

METS-IR has garnered significant interest in metabolic research. This index, calculated from plasma glucose, triglycerides, high-density lipoprotein cholesterol (HDL-c), and body mass index (BMI), offers a practical alternative to insulin measurement while maintaining a strong correlation with the gold-standard euglycemic-hyperinsulinemic clamp assessments [22,23].

Although BMI remains the predominant metric in clinical and epidemiological research, its limitations—most notably its inability to differentiate between adipose tissue and lean mass, as well as to assess fat distribution—are well documented. These constraints have spurred the development of advanced anthropometric indices designed to better encapsulate visceral adiposity and its related cardiometabolic risks. The BRI, derived from waist circumference and height using an elliptical model, has shown superior correlations with visceral fat volume, components of the metabolic syndrome, and insulin resistance across diverse cohorts. Studies further substantiate BRI’s improved predictive capacity, particularly its association with early metabolic dysregulation and insulin resistance in individuals with normative BMI [24,25,26,27].

The recent development of WWI, which normalizes waist circumference relative to body weight, has garnered attention due to its strong association with visceral adiposity, insulin resistance, and diabetes mellitus. National Health and Nutrition Examination Survey (NHANES) data indicate that each incremental increase in one unit in WWI correlates with elevated HOMA-IR levels and an increased long-term risk of all-cause and cardiovascular mortality in populations with prediabetes and diabetes. Moreover, WWI demonstrates superior predictive capacity over BMI and waist circumference for coronary artery disease prevalence in a younger demographic [24,27,28,29,30,31].

Other indices such as the A Body Shape Index (ABSI), Body Adiposity Index (BAI), Conicity Index (CI), and Sagittal Abdominal Diameter (SAD) provide nuanced insights into body composition that surpass traditional anthropometric measures. ABSI, which standardizes waist circumference in relation to height and weight, has demonstrated strong correlations with all-cause mortality and insulin resistance in males. Similarly, SAD is a robust surrogate marker for visceral adiposity and cardiovascular risk, functioning largely independently of BMI [25,27,32,33].

From a mechanistic standpoint, these composite IR scores underscore the interplay among key pathological pathways: lipotoxicity, glucotoxicity, and visceral adiposity. Elevated triglyceride levels promote ectopic fat accumulation and induce hepatic insulin resistance, while low high-density lipoprotein cholesterol (HDL-c) serves as a marker of compromised reverse cholesterol transport and heightened inflammatory states. Consequently, integrative indices that encompass lipid, glycemic, and adiposity parameters offer more accurate stratification capabilities compared to isolated biomarkers [34,35,36,37,38]. We proposed similar integrative approaches in chronic kidney disease, where panels of inflammatory, oxidative, and atherogenic biomarkers, including proprotein convertase subtilisin/kexin type 9 (PCSK9), soluble epoxide hydrolase (EPHX2), advanced oxidation protein products (AOPP), and thiobarbituric acid reactive substances (TBARS), better capture cardiorenal risk than single markers [39].

In light of the preceding considerations, this investigation aims to rigorously compare METS-IR and TyG-derived indices against established biomarkers such as HOMA-IR and fasting insulin, as well as advanced anthropometric measures including BRI and WWI, and examine their correlations with β-cell function, visceral adiposity, and cardiovascular risk markers. By contextualizing our findings within this evolving scientific landscape, we aim to address a significant gap in the validation and application of novel surrogate insulin resistance indices across cohorts with mixed glycemic status.

## 2. Materials and Methods

Our research was a non-interventional, cross-sectional epidemiological study conducted over a six-month period. The study was conducted in accordance with the Declaration of Helsinki and received ethical approval from the Ethics Committee of the Filantropia Municipal Clinical Hospital in Dolj, Romania (approval no. 886/15 January 2024).

### 2.1. Study Design and Population

This was a cross-sectional analytical study conducted on a total of 200 adult patients recruited from the Filantropia Municipal Clinical Hospital, Craiova, Dolj, Romania. All of our patients met the criteria in order to classify them as suffering from MetS. Participants were categorized into two groups based on ADA criteria: 80 individuals with PreDM and 120 with T2DM. Inclusion criteria included age > 18 years (upper age limit 85) and confirmed diagnosis of PreDM or T2DM based on fasting plasma glucose (FPG), 2 h post-load glucose (2 h-PG), and/or glycated hemoglobin (HbA1c). All of our patients were diagnosed with PreDM and T2DM. Exclusion criteria included type 1 diabetes, chronic liver or kidney disease, malignancy, and acute infections. Data were collected on glucose-lowering drug use.

The primary aim of the study was to collect comprehensive data across multiple health and lifestyle domains. Information was obtained using a structured interview questionnaire and included anthropometric measurements, clinical variables, laboratory test results, and detailed demographic and lifestyle characteristics.

Demographic data analyzed comprised age, sex, monthly household income, and level of educational attainment. Lifestyle and health-related factors assessed included self-reported history of tobacco and alcohol use, family history of hypertension, diabetes mellitus, and cardiovascular disease, as well as the average weekly duration of intentional moderate physical activity. The study also documented current pharmacological treatments. Several participants were undergoing antihypertensive therapy, including monotherapy with perindopril or amlodipine, or a combination of perindopril and indapamide. Statin use was also recorded, with atorvastatin and rosuvastatin being the prescribed agents. Regarding dietary management, the majority of participants had not received individualized nutrition plans from a healthcare professional.

### 2.2. Diagnostic Framework for Study Populations: PreDM, T2DM, MetS

PreDM and T2DM were assessed using ADA and IDF criteria [8,9]. Diagnosis of MetS followed the 2009 harmonized criteria, requiring at least three of five specified metabolic risk factors [40].

Of the initial 288 patients diagnosed with T2DM, 120 completed the study and were included in the final analysis. Similarly, 80 out of 189 patients from the PreDM group met the follow-up criteria and were included in the final evaluation. The 168 diabetic patients and 109 prediabetic patients were not retained in follow-up for the following reasons: relocation, reluctance to continue, patients who have been diagnosed with chronic microvascular complications (diabetic retinopathy, diabetic nephropathy, peripheral polyneuropathy), patients diagnosed with chronic infectious or inflammatory conditions, patients with any history of malignancy, those who had suffered from an acute infection or inflammatory disease in the previous month, anyone younger than eighteen years, and pregnant women (Figure 1).

### 2.3. Assessment of Anthropometric and Adiposity Indices

The following anthropometric indices were calculated [41,42,43,44,45,46,47,48]:Body Mass Index: BMI = Weight/(Height)^2^;Body Roundness Index: BRI = 364.2–365.5 × (1 − [WC/2π]^2^/[0.5 × height^2^)^½^;A Body Shape Index (ABSI): ABSI = WC/[BMI^2/3^ × Height^1/2^];A Body Shape Index z Score was determined with omniCalculator;Weight-Adjusted Waist Index: WWI = WC (cm)/√ Weight (kg);Abdominal Volume Index: AVI = [2 × WC^2^ + 0.7 × (WC − HC)^2^]/1000;Conicity Index: CI = WC (m)/[0.109 × √(Weight (kg)/Height (m))].

### 2.4. Assessment of Lipemic Spectrum Indices

The following lipidic indices were calculated [49,50,51,52,53,54,55,56]:Castelli Risk Index I: CRI I = TC/HDL-c;Castelli Risk Index II: CRI II = LDL-c/HDL-c;TyG Index: TyG = Ln [TG (mg/dL) × FPG (mg/dL)/2];TyG to HDL-c ratio = TyG/HDL-c;Lipoprotein Combine Index: LCI = TC × TG × LDL-c/HDL-c;Lipid adipose product: LAP  =  (WC − 65) × TG for men; LAP  =  (WC − 58) × TG for women;Triglyceride-total cholesterol-body weight index: TCBI = Triglyceride (mg/dL) × Total Cholesterol (mg/dL) × Body Weight (kg)/1000.

### 2.5. Assessment of Insulin Resistance

The following IR indices were calculated [57,58,59]:Homeostatic Model Assessment of Insulin Resistance: HOMA-IR = (Fasting Insulin [μUI/mL] × FPG [mg/dL])/405;β-cell Function Index: HOMA%B = (360 × Fasting Insulin (μUI/mL))/(FPG (mg/dL) − 63);Metabolic Score for Insulin Resistance: METS-IR = (ln((2 × FPG (mg/dL)) + TG (mg/dL)) × BMI (kg/m^2^))/(ln(HDL-c (mg/dL))).

### 2.6. Laboratory-Based Assays

In the course of venous blood sampling, approximately 5 mL of venous blood was collected from each subject using additive-free Becton Dickinson Vacutainer tubes (Franklin Lakes, NJ, USA). Following established laboratory protocols, the samples were allowed to clot and were subsequently centrifuged within four hours of collection at 3000× *g* for 10 min using a Hermle centrifuge (Hermle AG, Gosheim, Baden-Württemberg, Germany). The resulting serum was aliquoted into pre-labeled, airtight vials to reduce the risk of contamination and was stored at temperatures ranging from −20 °C to −80 °C to ensure stability. To maintain sample integrity, stringent precautions were taken to prevent freeze–thaw cycles. Prior to analysis, the serum specimens were permitted to thaw passively at ambient temperature. Peripheral venous blood samples were collected in ethylenediaminetetraacetic acid (EDTA)-containing Vacutainer tubes for subsequent complete blood count (CBC) analysis.

Following the collection of anthropometric measurements, participants underwent laboratory-based evaluations to further assess clinical and biochemical parameters. Laboratory analyses were conducted using the ARCHITECT C4000 clinical chemistry analyzer (Abbott, Abbott Park, IL, USA) to measure standard biochemical markers.

A CBC and differential white blood cell count were performed using the hematology analyzer MINDRAY BC-6800 (Mindray, Shenzhen, China).

### 2.7. Statistical Analysis

Data processing and initial organization were carried out using Microsoft Excel. Statistical analyses were performed using GraphPad Prism version 10.3.1 (GraphPad Software LLC., San Diego, CA, USA) and R version 4.3.2 to facilitate advanced modeling, correlation networks, and graphical visualization.

Normality of the data was assessed using the D’Agostino and Pearson omnibus test. Continuous variables following a Gaussian distribution were expressed as mean ± standard deviation (SD), while non-normally distributed variables were reported as median and interquartile range (IQR). One-way ANOVA was applied to compare means across multiple groups for normally distributed variables, whereas the Kruskal–Wallis test was used for non-Gaussian data. Categorical variables were analyzed using the χ^2^ (chi-square) test. To explore associations among variables, Spearman’s rank correlation coefficients were calculated. Statistical significance was defined as *p* < 0.05 across all tests.

## 3. Results

The analysis compared clinical, metabolic, and anthropometric indices across prediabetic and diabetic cohorts to evaluate their association with insulin resistance, β-cell function, and cardiovascular risk. The results are presented in five parts: baseline characteristics of the study population, stratified analysis based on HOMA-IR and METS-IR categories, correlations among the two cohorts, a comparison of insulin resistance and cardiometabolic markers according to hypertension grade, and multiple linear regression of different models.

### 3.1. Clinical and Biochemical Characteristics of Prediabetic and Diabetic Subjects

Table 1 presents the baseline clinical, anthropometric, and biochemical characteristics of the study population, comparing individuals with prediabetes (PreDM, *n* = 80) and type 2 diabetes mellitus (T2DM, *n* = 120).

Age was significantly higher in the T2DM group (64.17 ± 10.80 years) compared to the PreDM group (58.40 ± 11.79 years, *p* = 0.0005), consistent with the progressive nature of glucose dysregulation over time. Diabetic participants had markedly elevated glycemic parameters: HbA1c (median 9.09 vs. 5.90, *p* < 0.0001), FPG (161.00 vs. 100.50 mg/dL, *p* < 0.0001), and 2 h post-load glucose (249.50 vs. 158.50 mg/dL, *p* < 0.0001), confirming the diagnosis. TC and LDL-c levels were significantly lower in the diabetic group (*p* = 0.0002 and *p* = 0.0001, respectively), likely due to more frequent use of lipid-lowering therapies in established T2DM. HDL-c was significantly lower in T2DM (44.50 vs. 54.30 mg/dL, *p* = 0.0001), aligning with known patterns of diabetic dyslipidemia. AST and ALT were significantly elevated in the diabetic group (*p* = 0.011 and *p* = 0.004, respectively), suggesting higher hepatic stress or steatosis. CRP and WBC were significantly higher in T2DM, reflecting a pro-inflammatory state commonly associated with insulin resistance and vascular risk (*p* < 0.0001 and *p* = 0.033).

### 3.2. Comparison of Insulin Resistance and Anthropometric Indices Across HOMA-IR and METS-IR Categories in Prediabetic and Diabetic Groups

Table 2 and Table 3 stratify both PreDM and T2DM participants based on combinations of normal/altered HOMA-IR and METS-IR levels, offering insight into how these two insulin resistance indices relate to β-cell function and anthropometric markers. In order to stratify, we considered METS-IR and HOMA-IR normal for the values that were lower than their means, and altered for the values equal or higher (PreDM: METS-IR mean value of 42.76, HOMA-IR mean value of 2.86; T2DM: METS-IR mean value of 49.89, HOMA-IR mean value of 5.55).

#### 3.2.1. Prediabetic Group

Homeostasis Model Assessment of β-cell Function (HOMA%B) increased progressively from the normal HOMA-IR/normal METS-IR group (median 79.76) to the altered HOMA-IR/altered METS-IR group (median 185.70), with a significant difference across all subgroups (*p*< 0.0001). This suggests active β-cell compensation in response to insulin resistance in prediabetes. BMI and BRI also followed this pattern, being significantly higher in subgroups with altered METS-IR (*p* < 0.0001 and *p* = 0.0003, respectively), indicating stronger association of METS-IR with adiposity-related insulin resistance. ABSI Z-score varied significantly among groups (*p* = 0.007), with the highest values seen in those with altered HOMA-IR but normal METS-IR, and the lowest in those with altered METS-IR but normal HOMA-IR. This variation could suggest differing body shape contributions to insulin resistance depending on the index used. Indices like WWI, CI, and AVI were not significantly different among groups (except ABSI at *p* = 0.035), suggesting they may have lower discriminative power in early dysglycemia.

#### 3.2.2. Type 2 Diabetes Group

HOMA%B was significantly lower in individuals with altered METS-IR and/or HOMA-IR, indicating progressive β-cell failure typical of advanced disease (*p* < 0.0001). HbA1c rose across subgroups, peaking in those with both indices altered (*p* = 0.001), confirming worsening glycemic control in more insulin-resistant individuals. BMI, BRI, AVI, and WWI were all significantly elevated in subgroups with altered METS-IR, especially when both HOMA-IR and METS-IR were abnormal (*p* < 0.0001 for BMI, BRI, AVI; *p* = 0.0004 for WWI). This supports the idea that METS-IR more strongly captures adiposity-related insulin resistance compared to HOMA-IR. CI was modestly but significantly higher among those with altered METS-IR (*p* = 0.002), while ABSI and ABSI Z-score did not differ significantly among diabetic subgroups, in contrast to the prediabetic group. This may reflect the ceiling effect of ABSI once diabetes is fully established.

### 3.3. Comparison of Insulin Resistance and Cardiometabolic Indices by Hypertension Grade in Prediabetic and Diabetic Individuals

Table 4 evaluates key IR and cardiometabolic indices in individuals with Grade II and Grade III hypertension, comparing PreDM and T2DM subgroups. This analysis helps elucidate how hypertension severity influences or coexists with metabolic derangements in each glycemic state.

Across both hypertension grades, HOMA-IR was significantly higher in the diabetic group (Grade II: 4.33 vs. 2.66, *p* = 0.022; Grade III: 4.32 vs. 2.49, *p* < 0.0001), indicating greater insulin resistance in T2DM independent of blood pressure severity. HOMA%B, a marker of β-cell function, was significantly lower in T2DM for both HTN grades (Grade II: 46.74 vs. 101.90; Grade III: 40.83 vs. 111.00; both *p* < 0.0001), reinforcing the expected β-cell dysfunction in diabetes. METS-IR, a non-insulin-based IR index, was also significantly elevated in diabetics for both Grade II (46.46 vs. 39.88, *p* = 0.015) and Grade III HTN (51.04 vs. 44.49, *p* = 0.0009), confirming its robustness in detecting IR across varying BP levels. TyG index, a widely used surrogate for IR, was significantly higher in T2DM across both grades (*p* < 0.0001), further supporting elevated lipid-related metabolic burden in diabetes. TyG/HDL-c ratio was also significantly elevated in diabetics (Grade II: *p* = 0.016; Grade III: *p* < 0.0001), reinforcing its utility as a simple, sensitive IR marker. TG/HDL-c ratio was significantly different in Grade II HTN (*p* = 0.049), but not in Grade III, suggesting this ratio may lose discriminatory power as metabolic disease advances.

### 3.4. Correlations Among PreDM and T2DM Cohorts

#### 3.4.1. Correlations of Prediabetic and Diabetic Individuals

Figure 2 presents the heatmaps of several correlations for both PreDM and T2DM cohorts.

The most significant correlations are identified as follows:

PreDM cohort:

METS-IR correlated positively and moderately with BRI (*r* = 0.480), and TCBI (*r* = 0.510);WWI correlated positively and strongly with ABSI (*r* = 0.860) and CI (*r* = 0.940);LAP correlated positively and moderately with AVI (*r* = 0.580);

T2DM cohort:

METS-IR correlated positively and strongly with BRI (*r* = 0.800) and AVI (*r* = 0.730);WWI correlated positively and strongly with BRI (*r* = 0.830), ABSI (*r* = 0.820) and CI (*r* = 0.950);LAP correlated positively and strongly with AVI (*r* = 0.764).

#### 3.4.2. Correlation of PreDM and HTN Grade 2 Cohort; Correlation of PreDM and HTN Grade 3 Cohort

We constructed heat map correlation graph based on all numeric variables for PreDM and HTN Grade 2 cohort and T2DM and HTN Grade 2 cohort, as can be seen in Figure 3.

##### PreDM and HTN Grade 2 Cohort

In individuals with prediabetes and hypertension grade 2, several strong positive correlations (*r* ≥ 0.70) were observed among anthropometric and cardiometabolic indices. Similarly, ABSIzScore showed a near-perfect correlation CI (*r* = 0.99), while WWI correlated strongly with both CI (*r* = 0.99) and ABSI (*r* = 0.97).

Obviously, BMI was also highly correlated with METS-IR (*r* = 0.95), underscoring the metabolic link between obesity and insulin resistance in this cohort.

Among lipid-derived markers, CRI I and CRI II were correlated with the LCI (*r* = 0.83 and *r* = 0.80, respectively). Additionally, TG/HDLc ratio was strongly correlated with the TyG index (*r* = 0.81), as well as with TCBI (*r* = 0.75), suggesting consistent associations between triglyceride-based markers.

These interrelationships highlight the clustering of cardiometabolic risk factors in prediabetic individuals with moderate hypertension and support the use of integrated indices in risk stratification.

##### DM and HTN Grade 2 Cohort

Among individuals with type 2 diabetes and hypertension grade 2, several robust correlations (*r* ≥ 0.70) were observed between indices of adiposity, insulin resistance, and cardiometabolic risk. WWI showed a very strong relationship with CI (*r* = 0.98), and ABSIzScore also strongly correlated with CI (*r* = 0.98) and ABSI (*r* = 0.97), reinforcing the consistency of abdominal adiposity markers.

Other obvious correlations included BRI with AVI (*r* = 0.96) and WWI with ABSIzScore (*r* = 0.96). CRI I and LCI shared a strong association (*r* = 0.94), while TG/HDLc correlated with TyG/HDLc (*r* = 0.90), suggesting similar predictive properties.

Consistent clustering was also observed between BMI and METS-IR (*r* = 0.84), as well as with BRI (*r* = 0.82) and AVI (*r* = 0.78), affirming obesity’s close ties to insulin resistance and abdominal geometry in this group. Additionally, TyG showed strong correlations with TG/HDLc (*r* = 0.82) and TCBI (*r* = 0.73), providing further insight into lipid-insulin dynamics.

#### 3.4.3. Correlation of PreDM and HTN Grade 3 Cohort; Correlation of T2DM and HTN Grade 3 Cohort

We constructed heat map correlation graph based on all numeric variables for PreDM and HTN Grade 3 cohort and T2DM and HTN Grade 3 cohort, as can be seen in Figure 4.

##### PreDM and HTN Grade 3 Cohort

In the prediabetic individuals with grade 3 hypertension, multiple strong correlations (*r* ≥ 0.70) were detected among anthropometric, metabolic, and lipid-derived indices. ABSIzScore was highly correlated with the CI (*r* = 0.98), indicating close interrelationships among central adiposity metrics.

WWI demonstrated robust correlations with CI (*r* = 0.97), ABSIzScore (*r* = 0.93), and ABSI (*r* = 0.90), again underscoring the coherence between various waist-related indicators. Additionally, TG/HDLc showed strong correlations with both LAP (*r* = 0.91) and the TyG index (*r* = 0.84), confirming the clinical interdependence of triglyceride-based risk factors.

Furthermore, CRI I and II were significantly associated with LCI (*r* = 0.85 and *r* = 0.73, respectively). BMI correlated strongly with METS-IR (*r* = 0.90), highlighting the persistent link between obesity and metabolic dysfunction in this high-risk group.

These findings emphasize an even more pronounced clustering of risk markers in individuals with more severe hypertension, which could have implications for early intervention and integrated risk assessment.

##### DM and HTN Grade 3 Cohort

In patients with type 2 diabetes and grade 3 hypertension, numerous strong positive correlations (*r* ≥ 0.70) were observed, underscoring tight interconnections among anthropometric, metabolic, and lipid-related indices. The WWI exhibited exceptionally strong correlations with CI (*r* = 0.97), ABSIzScore (*r* = 0.90), and ABSI (*r* = 0.85), confirming its consistency as a central obesity marker. Similarly, ABSIzScore and CI were highly correlated (*r* = 0.96).

BRI correlated strongly with both AVI (*r* = 0.93) and CI (*r* = 0.77), while BMI and METS-IR shared a close relationship (*r* = 0.92), further affirming obesity’s central role in metabolic impairment.

Cardiometabolic indices also clustered together, with CRI I correlating with LCI (*r* = 0.84), while TG/HDLc was closely linked to both TCBI (*r* = 0.85) and LAP (*r* = 0.80). TyG showed substantial correlations with TCBI (*r* = 0.80) and TG/HDLc (*r* = 0.78), emphasizing the synergistic interplay between lipid markers and insulin resistance.

### 3.5. Multiple Linear Regression (MLR)

#### 3.5.1. MLR of PreDM Group

In the prediabetes group, we fitted two multiple linear regression models with METS-IR-PreDM as the dependent variable, using data from 80 participants and four parameters per model (Table 5).

In Model 1, METS-IR-PreDM was regressed on HOMA-IR-PreDM, TyG-PreDM and BRI-PreDM. The model was statistically significant overall, F(3,76) = 6.11, *p* = 0.0009, but explained a modest proportion of the variance, R^2^ = 0.1943. TyG-PreDM and BRI-PreDM showed independent associations with METS-IR-PreDM, with β = 7.480, 95% CI 3.513 to 11.45, *p* = 0.0003 for TyG-PreDM and β = 1.288, 95% CI 0.318 to 2.258, *p* = 0.0100 for BRI-PreDM, whereas HOMA-IR-PreDM was not significant, F(1,76) = 0.002, *p* = 0.9629. The intercept was negative, β_0_ = −30.54, 95% CI −66.69 to 5.62. Variance inflation factors for the three predictors were close to 1, between 1.02 and 1.05, indicating no relevant multicollinearity. Residual normality tests were mixed, with two of the four tests failing at alpha 0.05, suggesting some deviation from normality of residuals.

Model 2 included HOMA%B-PreDM, LAP-PreDM and WWI-PreDM as predictors. This model showed a better fit, with a higher explained variance, R^2^ = 0.3139, and a highly significant global F test, F(3,76) = 11.59, *p* < 0.0001. Among the predictors, LAP-PreDM and WWI-PreDM were strongly and independently associated with METS-IR-PreDM, with β = 0.0749, 95% CI 0.0399 to 0.1099, *p* < 0.0001 for LAP-PreDM and β = −2.501, 95% CI −3.539 to −1.463, *p* < 0.0001 for WWI-PreDM, while HOMA%B-PreDM was not significant, F(1,76) = 0.259, *p* = 0.6123. The intercept was positive, β_0_ = 66.17, 95% CI 53.99 to 78.35. As in Model 1, multicollinearity was negligible, with VIF values between 1.04 and 1.07. For Model 2, three of the four normality tests were passed, with only the D’Agostino–Pearson omnibus test remaining significant, indicating a mild but acceptable departure from perfect normality of residuals.

Formal comparison based on Akaike Information Criterion (AICc) clearly favored Model 2, with an AICc difference of 12.86, a probability of 99.84 percent that Model 2 is the better model versus 0.16 percent for Model 1, and a probability ratio of 620.7.In this cohort with prediabetes, METS-IR-PreDM was therefore described more accurately by the model including LAP-PreDM and WWI-PreDM, while HOMA-derived indices contributed little independent information.

#### 3.5.2. MLR of T2DM Group

We performed multiple linear regression analyses to evaluate the association between METS-IR-DM and two different sets of predictors in 120 patients with diabetes (Table 6).

In Model 1, METS-IR-DM was regressed on HOMA-IR-DM, TyG-DM and BRI-DM. This model showed a good fit, explaining 68.9% of the variability in METS-IR-DM (R^2^ = 0.6888, DF = 116), with a highly significant overall F test, F(3, 116) = 85.59, *p* < 0.0001. TyG-DM and BRI-DM were both independently and positively associated with METS-IR-DM, with β = 5.947, 95% CI 3.922 to 7.973, *p* < 0.0001, and β = 3.316, 95% CI 2.792 to 3.839, *p* < 0.0001, respectively. HOMA-IR-DM was not an independent predictor in this model, β = −0.072, *p* = 0.4543. The intercept was negative and statistically significant, β_0_ = −23.41, *p* = 0.0124. Variance inflation factors were low, between 1.04 and 1.11, indicating no relevant multicollinearity. Residual normality tests, including D’Agostino Pearson, Anderson Darling, Shapiro–Wilk and Kolmogorov–Smirnov, all had *p* values above 0.05, supporting an approximately normal distribution of residuals and adequate model assumptions.

In Model 2, METS-IR-DM was regressed on HOMA%B-DM, LAP-DM and WWI-DM. This model explained less of the outcome variability, with R^2^ = 0.4978 (DF = 116), although the overall F test remained significant, F(3, 116) = 38.33, *p* < 0.0001. Among the predictors, only LAP-DM showed a significant positive association with METS-IR-DM, β = 0.1124, 95% CI 0.0855 to 0.1392, *p* < 0.0001. HOMA%B-DM and WWI-DM were not significant, with *p* = 0.2246 and *p* = 0.1438, respectively. The intercept was positive and significant, β_0_ = 31.79, *p* < 0.0001. VIF values ranged from 1.03 to 1.32, again excluding important multicollinearity. Residual normality was acceptable in most tests, with a slight deviation suggested by one test, without a clear impact on the overall interpretation.

Formal model comparison based on AICc strongly favored Model 1. The AICc difference was −57.43 in favor of Model 1, with an estimated probability above 99.99 percent that Model 1 is the better model, and a likelihood ratio of about 2.95 × 10^12^ relative to Model 2. Therefore, the model including TyG-DM and BRI-DM as main predictors was selected as the preferred model to describe METS-IR-DM in this cohort.

## 4. Discussion

In this study, we compared insulin resistance and metabolic dysfunction markers between PreDM and T2DM individuals, focusing on anthropometric, biochemical, and derived indices stratified by hypertension status. Our results extend current evidence in clinically relevant ways.

T2DM patients demonstrated significantly worse glycemic control, altered lipid parameters (particularly lower HDL-c), elevated liver enzymes, and heightened inflammation compared to their prediabetic counterparts. In Romanian cohorts with PreDM and newly diagnosed T2DM, we previously demonstrated that pentraxin 3 along with other inflammatory biomarkers are already elevated and show correlations with various obesity indices, reinforcing the idea that low-grade inflammation is associated with early dysglycemia [11].

Surprisingly, some classic cardiometabolic markers (BMI, TG, and creatinine) were not significantly different between groups, suggesting that metabolic alterations in T2DM are not always reflected by traditional anthropometric or renal markers.

In PreDM, increased HOMA%B and elevated anthropometric indices reflect an active compensatory phase in insulin resistance, particularly when METS-IR is elevated, suggesting METS-IR is sensitive to early metabolic stress. In T2DM, insulin resistance correlates with significantly lower β-cell function and higher adiposity indices, particularly BMI, BRI, and AVI, again with METS-IR identifying more severe metabolic profiles than HOMA-IR alone. Notably, ABSI Z-score may serve as a more sensitive marker in the prediabetic stage but loses significance in overt diabetes.

HOMA-IR, METS-IR, TyG, and TyG/HDL-c consistently distinguish between PreDM and T2DM even when hypertension is present, underscoring their robustness across clinical stages. Traditional atherogenic and lipid accumulation indices (CRI I/II, LAP, LCI, TCBI) were less effective in distinguishing metabolic risk in this hypertensive population. These findings suggest that triglyceride-glucose-derived indices (especially TyG and TyG/HDL-c) and insulin-based models remain reliable markers of worsening metabolic risk in hypertensive individuals, regardless of glycemic stage.

Our findings indicate a heightened degree of metabolic clustering in patients with diabetes and severe hypertension, supporting the utility of composite indices for comprehensive risk profiling in high-burden populations.

The present study revealed that both HOMA-IR and METS-IR are significantly elevated in individuals with T2DM compared to those with PreDM, consistent with established pathophysiological progression. Our findings are aligned with recent studies that confirm HOMA-IR’s continued validity in reflecting insulin resistance across glycemic spectrums. For instance, it was demonstrated that HOMA-IR remains a superior predictor of insulin resistance and hepatic steatosis in individuals with T2DM, while METS-IR showed better performance in early metabolic derangements [60,61,62].

Interestingly, while both HOMA-IR and METS-IR were elevated in T2DM, the overlap observed in METS-IR distributions between groups suggests this index is more sensitive to mixed metabolic states than sharply discriminative. This mirrors current literature, which found METS-IR better suited for identifying insulin resistance in normoglycemic or borderline populations than in overt T2DM [38,63,64,65,66].

Our results also emphasized the utility of anthropometric markers like BRI, WWI, and ABSI in capturing body fat distribution, particularly in relation to METS-IR. Consistent with our findings, BRI and WWI were highlighted as reliable predictors of visceral adiposity and metabolic syndrome risk, with stronger associations in diabetic than prediabetic populations. ABSI loses predictive value in advanced T2DM due to body shape adaptations and plateauing of abdominal fat indices [31,67,68,69].

The network analysis from our study revealed central roles for insulin-related indices and adiposity markers. This interconnected pattern echoes findings from the NHANES mortality study, in which METS-IR emerged as a key integrative marker of cardiometabolic dysfunction [35].

In hypertensive subgroups, our study confirmed that TyG and TyG/HDL-c remain consistent markers for metabolic burden, even under elevated blood pressure. Our findings in hypertensive subgroups align well with large-scale epidemiological evidence. A 2023 meta-analysis showed that each 1-unit increase in TyG index raised hypertension risk by 1.5-fold. Moreover, an NHANES-based machine learning study confirmed that TyG (with TyG–WHtR) independently predicts mortality in hypertensive adults [69,70,71,72].

Despite the strengths of multiple indices, our results suggest that a composite approach using: HbA1c, TyG, and either HOMA-IR or METS-IR, offers the best overall performance in characterizing insulin resistance across diverse clinical stages. This echoes calls from current literature for integrated diagnostic models, suggesting the combination anthropometric, glycemic, and lipid markers for precision metabolic screening [73,74].

In the context of β-cell function, our data highlighted a marked reduction in HOMA%B in individuals with T2DM, particularly among those with elevated HOMA-IR and METS-IR indices. This supports the widely accepted model of β-cell exhaustion following chronic compensatory hyperinsulinemia. Literature findings confirm that a progressive decline in HOMA%B begins in the late prediabetic phase and sharply accelerates after overt diabetes onset, especially in individuals with concurrent obesity and dyslipidemia [75]. Our subgroup analysis revealed that in PreDM, β-cell compensation (reflected by high HOMA%B) occurs in the presence of altered METS-IR, highlighting the latter’s sensitivity to metabolic stress even before glycemic markers fully deteriorate.

Moreover, the limited utility of composite indices such as LAP, CRI I/II, and LCI in distinguishing between PreDM and T2DM within hypertensive subpopulations warrants attention. While these markers have shown utility in normotensive cohorts, their predictive performance appears diminished under the physiological confounding introduced by elevated blood pressure and vascular remodeling [76,77,78,79].

Of particular interest is the distinct behavior of METS-IR as a non-insulin-based surrogate that still correlates significantly with HOMA-IR and β-cell decline in T2DM. METS-IR incorporates routine clinical parameters: fasting glucose, triglycerides, and HDL-c, making it not only practical but also broadly applicable across different healthcare settings. Its inverse association with HOMA-IR in advanced diabetes, observed in our study and supported by recent data, may reflect a paradoxical shift where visceral adiposity remains high but insulin secretion declines due to islet failure [35,80,81,82,83,84,85].

Despite the strong performance of METS-IR and TyG in detecting insulin resistance across prediabetes and T2DM, their integration into routine clinical workflows remains underexplored. For instance, in patients with prediabetes or hypertension, METS-IR values approaching or exceeding 40 and TyG values above 8.8 were associated with greater insulin resistance and cardiometabolic burden in our study. These thresholds align with those reported in previous population-based research and may help guide early lifestyle interventions or closer metabolic follow-up [84,86]. These indices are particularly useful when insulin assays are unavailable, making them valuable tools in primary care and resource-limited settings. Additionally, their ability to reflect changes in metabolic status supports their use in continuous monitoring, complementing HbA1c and BMI. Future research should validate cut-off thresholds and evaluate the effectiveness of including these indices in routine screening and decision-making protocols.

In summary, our findings provide robust support for the clinical relevance of METS-IR, TyG, and advanced anthropometric indices in evaluating insulin resistance and β-cell function, particularly in the context of T2DM and hypertension. Our findings reinforce the utility of METS-IR and TyG (including its HDL-c-adjusted variant) as robust markers of insulin resistance and metabolic risk across diabetic and hypertensive contexts. HOMA-IR remains valuable when insulin measurements are available, but METS-IR’s broader applicability, validated in multiple large cohorts, positions it as a versatile tool in both research and clinical screening. Strategic integration of these indices can aid early detection, risk stratification, and targeted intervention in PreDM and T2DM patients. The distinct profiles and predictive power of these indices across different stages of glucose intolerance reinforce the necessity of stage-specific diagnostic strategies for optimal metabolic risk stratification.

### Limitations

Despite the valuable insights offered by this study, several limitations should be acknowledged. Firstly, its cross-sectional design restricts the ability to establish temporal or causal relationships between insulin resistance markers and the progression from prediabetes to T2DM. Longitudinal data would be more suitable to capture dynamic metabolic changes over time.

Secondly, the study cohort was drawn from a single tertiary care center in Romania, which may introduce selection bias and limit the generalizability of the findings to broader or more diverse populations, particularly those with different ethnic backgrounds or healthcare access. Our findings should be interpreted in the context of a Romanian cohort, and future studies comparing similar populations across Central and Eastern Europe would help clarify whether these results are consistent in regions with shared economic and lifestyle characteristics.

Furthermore, while the study incorporated a wide range of anthropometric and lipid-based indices, some key variables such as physical activity level, dietary intake, and socioeconomic status were not assessed, despite their known influence on metabolic health.

Crucially, the sample size, particularly within stratified subgroups such as altered METS-IR or combined hypertensive-diabetic categories, was relatively small, which may have limited the statistical power to detect subtle differences or interactions between variables. Larger, more representative cohorts are needed to confirm these findings and enhance the reliability and external validity of the conclusions. By design, we excluded patients with chronic kidney disease or established microvascular complications, although in our previous research on diabetic nephropathy and hemodialysis cohorts, protein oxidation markers such as AOPP were closely linked to inflammatory markers and identified as useful candidates for monitoring diabetic kidney disease [87].

All participants in this study met the criteria for metabolic syndrome, reflecting an already advanced cardiometabolic state. This inclusion criterion limits the generalizability of our findings to the broader populations with PreDM or T2DM who do not have MetS. Hypertension, which was prevalent across all subgroups, was not fully controlled in the statistical models, and this may have influenced the observed associations between insulin resistance indices and anthropometric markers.

## 5. Conclusions

In adults with MetS and impaired glucose regulation, both insulin-based and non-insulin-based indices of insulin resistance increased with worsening metabolic profile. However, METS-IR and TyG, alone and combined with TyG/HDL c, showed stronger and more consistent associations with anthropometric and cardiometabolic markers than HOMA-IR, especially in T2DM and higher hypertension grades.

Anthropometric indices that capture central and visceral adiposity, particularly BRI, AVI, WWI and LAP, clustered closely with METS-IR, TyG and lipid risk ratios, in both PreDM and T2DM and in hypertensive subgroups. Traditional lipid ratios, such as TG/HDL c and Castelli indices, were less discriminative in advanced cardiometabolic disease.

These findings support the use of METS-IR and TyG, together with simple anthropometric measures like BRI, AVI and WWI, as practical tools for insulin resistance assessment in clinical settings where insulin measurements are unavailable, unreliable or costly. In patients with PreDM, T2DM and hypertension, these composite indices may help refine cardiometabolic risk stratification and guide earlier, more targeted intervention. Further longitudinal studies are needed to test their prognostic value for hard outcomes and disease progression.

## Figures and Tables

**Figure 1 life-15-01845-f001:**
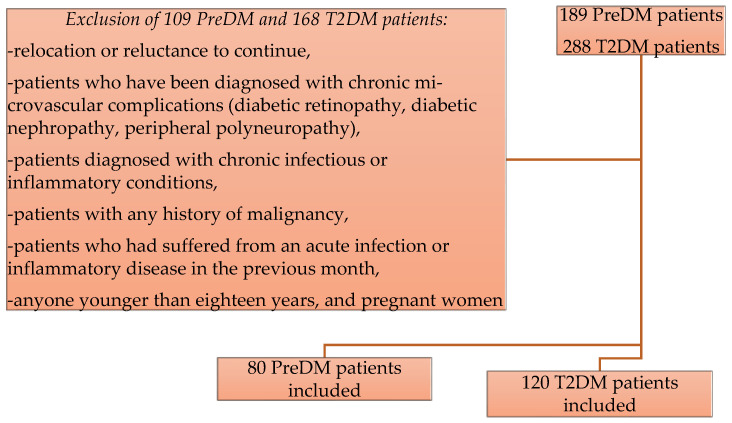
Diagram of the patients included in the study.

**Figure 2 life-15-01845-f002:**
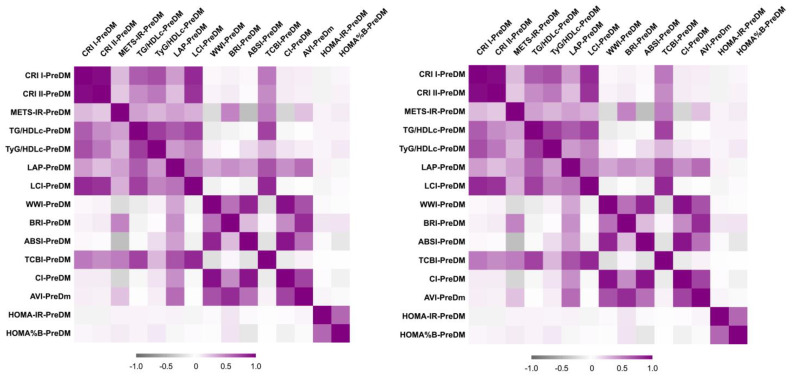
Correlation Heatmap of PreDM cohort (**left**) and T2DM cohort (**right**).

**Figure 3 life-15-01845-f003:**
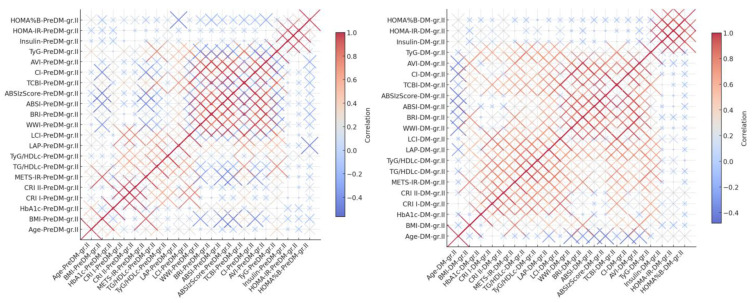
Correlation Heatmap-PreDM-HTNGr.II (**left**). Correlation Heatmap—T2DM-HTNGr.II (**right**).

**Figure 4 life-15-01845-f004:**
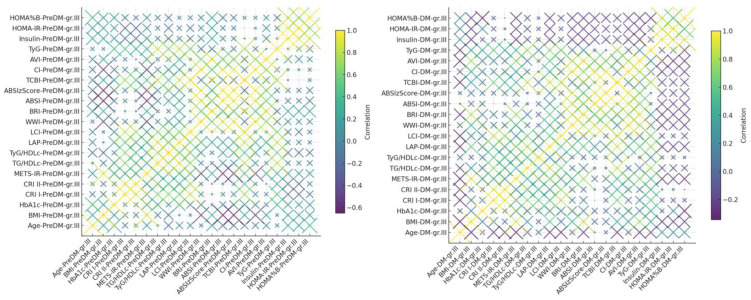
Correlation Heatmap–PreDM-HTNGr.III (**left**). Correlation Heatmap–T2DM-HTNGr.III (**right**).

**Table 1 life-15-01845-t001:** General Characteristics of the Study Cohorts.

Variables	PreDM (*n* = 80)	T2DM (*n* = 120)	*p*-Value	Variables	PreDM (*n* = 80)	T2DM (*n* = 120)	*p*-Value
Age (years) (Mean ± SD)	58.40 ± 11.79	64.17 ± 10.80	0.0005 *	AST/ALT [median(range)]	1.03 (0.47–2.70)	0.98 (0.35–3.89)	0.11
SBP (mmHg) (Mean ± SD)	137.90 ± 18.76	137.50 ± 18.10	0.88	CREATININE (mg/dL) [median(range)]	0.76 (0.36–1.75)	0.81 (0.40–7.42)	0.932
DBP (mmHg) (Mean ± SD)	79.91 ± 14.68	80.52 ± 10.97	0.73	CRP (mg/dL) [median(range)]	0.51 (0.10–12.80)	8.50 (3.20–48.50)	<0.0001 *
BMI (kg/m^2^) [median(range)]	29.17 (16.72–41.76)	29.37 (18.21–46.84)	0.159	ESR (mm/1st hour) [median(range)]	27.50 (5.00–115.00)	30.00 (4.00–140.00)	0.466
Insulin (μIU/mL) [median(range)]	10.35 (1.60–52.80)	11.60 (0.50–131.80)	0.874	WBC (×10^3^/μL) [median(range)]	7.34 (3.67–12.66)	7.95 (4.54–18.74)	0.033 *
HbA1c (%) [median(range)]	5.90 (4.70–6.70)	9.09 (5.40–15.50)	<0.0001 *	HGB (g/dL) [median(range)]	12.75 (7.70–17.20)	13.30 (8.10–20.00)	0.167
FPG (mg/dL) [median(range)]	100.50 (56.00–122.00)	161.00 (110.00–273.00)	<0.0001 *	PLT (×10^3^/μL) [median(range)]	261.00 (116.00–513.00)	240.50 (116.00–573.00)	0.575
2 h-PG (mg/dL) [median(range)]	158.50 (141.00–196.00)	249.50 (147.00–475.00)	<0.0001 *	Gender male/female (*n*)	40/40	59/61	0.511
TC (mg/dL) [median(range)]	205.50 (130.00–464.60)	182.50 (86.00–365.00)	0.0002	Residence urban/rural (*n*)	46/34	69/51	0.558
LDL-c (mg/dL) [median(range)]	119.50 (58.00–368.00)	100.00 (26.20–261.00)	0.0001	Dyslipidemia *n* (%)	69 (86%)	106 (88%)	0.409
HDL-c (mg/dL) [median(range)]	54.30 (32.00–86.14)	44.50 (21.00–84.00)	0.0001	AST (IU/L) [median(range)]	19.14 (9.85–75.08)	22.59 (10.72–146.20)	0.011 *
TG (mg/dL) [median(range)]	133.00 (45.00–610.00)	151.50 (53.00–624.00)	0.343	ALT (IU/L) [median(range)]	18.50 (7.00–90.00)	23.50 (7.00–261.00)	0.004 *
Alcohol consumption *n* (%)	38 (47%)	54 (45%)	0.419	Smoking *n* (%)	44 (55%)	72 (60%)	0.288

SBP—Systolic Blood Pressure; DBP—Diastolic Blood Pressure; BMI—Body Mass Index; FPG—Fasting Plasma Glucose; 2 h-PG—2-Hour Post-Load Glucose; TC—Total Cholesterol; LDL-c—Low-Density Lipoprotein Cholesterol; HDL-c—High-Density Lipoprotein Cholesterol; TG—Triglycerides; AST—Aspartate Aminotransferase; ALT—Alanine Aminotransferase; CRP—C-Reactive Protein; ESR—Erythrocyte Sedimentation Rate; WBC—White Blood Cell Count; HGB—Hemoglobin; PLT—Platelet Count; * *p* < 0.05: statistically significant. *p*-value from Pearson’s Chi-Squared/Student’s *t*-Test/Mann–Whitney Test.

**Table 2 life-15-01845-t002:** Comparison of Insulin Resistance Indices Based on HOMA-IR and METS-IR Stratification in PreDM cohort.

	PreDM				
	Normal HOMA-IR and METS-IR (*n* = 26)	Normal HOMA-IR and Altered METS-IR (*n* = 21)	Altered HOMA-IR and Normal METS-IR (*n* = 15)	Altered HOMA-IR and METS-IR (*n* = 18)	*p*-Value from Ordinary One-Way ANOVA/Kruskal–Wallis Test
HOMA%B [median(range)]	79.76 (−406.30–261.80)	87.30 (−1704.00–333.80)	143.10 (106.20–532.80)	185.70 (66.86–704.00)	<0.0001 *
HbA1c (%) [median(range)]	5.90 (4.70–6.70)	5.90 (5.00–6.30)	5.80 (4.70–6.40)	5.86 (4.90–6.30)	0.831
BMI [median(range)]	24.90 (16.72–33.70)	31.85 (24.96–58.48)	24.90 (18.01–29.34)	33.87 (28.40–46.41)	<0.0001 *
WWI [median(range)]	11.55 (7.14–17.24)	11.30 (5.99–13.13)	11.62 (9.42–20.74)	11.06 (10.40–12.65)	0.335
BRI [median(range)]	4.91 (0.84–13.50)	6.54 (2.61–9.20)	5.10 (2.13–15.83)	6.18 (5.26–9.20)	0.0003 *
CI [median(range)]	1.37 (0.86–2.14)	1.32 (0.68–1.59)	1.42 (1.13–2.55)	1.30 (1.18–1.45)	0.248
AVI [median(range)]	19.42 (7.90–38.13)	22.12 (10.17–34.05)	18.84 (10.66–61.62)	22.31 (17.40–34.05)	0.153
ABSI [median(range)]	0.09 (0.06–0.15)	0.08 (0.04–0.10)	0.09 (0.07–0.17)	0.08 (0.07–0.09)	0.035 *
ABSI Z score (Mean ± SD)	0.99 ± 1.83	0.12 ± 1.16	1.75 ± 2.60	−0.03 ± 0.51	0.007 *

HOMA%B—Homeostasis Model Assessment of β-cell Function; HbA1c—glycosylated hemoglobin; BMI—Body Mass Index; WWI—Weight-Adjusted Waist Index; BRI—Body Roundness Index; CI—Conicity Index; AVI—Abdominal Volume Index; ABSI—A Body Shape Index; ABSI Z Score—A Body Shape Index z-Score; * *p* < 0.05: statistically significant.

**Table 3 life-15-01845-t003:** Comparison of Insulin Resistance Indices Based on HOMA-IR and METS-IR Stratification in T2DM cohort.

	T2DM				
	Normal HOMA-IR and METS-IR (*n* = 45)	Normal HOMA-IR and Altered METS-IR (*n* = 36)	Altered HOMA-IR and Normal METS-IR (*n* = 27)	Altered HOMA-IR and METS-IR (*n* = 12)	*p*-Value from Ordinary One-Way ANOVA/Kruskal–Wallis Test
HOMA%B [median(range)]	40.54 (2.40–103.10)	22.94 (8.42–86.18)	68.19 (21.64–370.70)	45.12 (15.77–148.70)	<0.0001 *
HbA1c (%) [median(range)]	8.70 (6.30–13.41)	9.00 (6.10–15.50)	9.30 (5.40–11.73)	10.60 (7.50–15.50)	0.001 *
BMI [median(range)]	25.39 (18.21–34.26)	28.20 (28.20–46.84)	27.36 (21.30–32.81)	33.79 (28.41–40.16)	<0.0001 *
WWI [median(range)]	10.40 (7.95–14.03)	11.22 (10.00–12.64)	10.29 (8.19–12.19)	11.62 (10.42–12.52)	0.0004 *
BRI [median(range)]	4.26 (1.28–7.91)	7.29 (4.99–12.43)	3.94 (1.36–7.73)	6.65 (5.64–9.58)	<0.0001 *
CI [median(range)]	1.24 (0.88–1.70)	1.32 (1.19–1.47)	1.23 (0.94–1.48)	1.34 (1.20–1.45)	0.002 *
AVI [median(range)]	17.33 (7.37–31.25)	25.59 (17.69–35.43)	16.21 (9.00–24.65)	23.76 (19.72–32.26)	<0.0001 *
ABSI [median(range)]	0.08 (0.06–0.11)	0.08 (0.07–0.09)	0.08 (0.06–0.09)	0.08 (0.07–0.09)	0.458
ABSI Z score (Mean ± SD)	0.83 ± 1.23	1.13 ± 0.57	0.74 ± 1.08	1.24 ± 0.66	0.272

HOMA%B—Homeostasis Model Assessment of β-cell Function; HbA1c—glycosylated hemoglobin; BMI—Body Mass Index; WWI—Weight-Adjusted Waist Index; BRI—Body Roundness Index; CI—Conicity Index; AVI—Abdominal Volume Index; ABSI—A Body Shape Index; ABSI Z Score—A Body Shape Index z-Score; * *p* < 0.05: statistically significant.

**Table 4 life-15-01845-t004:** Analysis of IR and Cardiometabolic Indices by Hypertension Stage in PreDM and T2DM cohorts.

	PreDM	T2DM	*p*-Value from Student’s *t*-Test/ Mann–Whitney Test	PreDM	T2DM	*p*-Value from Student’s *t*-Test/ Mann–Whitney Test
Variables	HTN gr. II (*n* = 30)	HTN gr. II (*n* = 30)		HTN gr. III (*n* = 50)	HTN gr. III (*n* = 90)	
HOMA-IR [median(range)]	2.66 (0.80–7.76)	4.33 (1.25–62.10)	0.022 *	2.49 (0.42–11.73)	4.32 (0.17–18.66)	<0.0001 *
HOMA%B [median(range)]	101.90 (−1704.00–492.00)	46.74 (12.57–370.70)	0.0001 *	111.00 (13.09–704.00)	40.83 (2.40–342.60)	<0.0001 *
METS-IR (Mean ± SD)	39.88 ± 9.52	46.46 ± 10.78	0.015 *	44.49 ± 10.24	51.04 ± 11.35	0.0009 *
TyG (Mean ± SD)	8.72 ± 0.45	9.38 ± 0.61	<0.0001 *	8.80 ± 0.58	9.33 ± 0.60	<0.0001 *
TyG/HDL-c [median(range)]	0.16 (0.09–0.24)	0.19 (0.10–0.48)	0.016 *	0.16 (0.09–0.32)	0.21 (0.11–0.39)	<0.0001 *
TG/HDL-c [median(range)]	2.61 (0.59–6.02)	3.57 (0.80–15.22)	0.049 *	2.51 (0.60–19.06)	3.06 (1.06–17.83)	0.179
CRI I [median(range)]	3.70 (2.28–8.24)	3.71 (2.33–13.17)	0.840	3.84 (2.07–9.34)	3.79 (2.06–9.13)	0.997
CRI II [median(range)]	2.17 (1.01–6.53)	1.98 (0.60–5.43)	0.727	2.34 (1.02–6.54)	2.21 (0.76–6.53)	0.491
LAP [median(range)]	62.05 (4.83–190.60)	67.15 (−5.65–244.60)	0.900	57.26 (0.00–328.50)	60.18 (0.00–345.20)	0.565
LCI [median(range)]	64,233.00 (17,610.00–279,581.00)	54,906.00 (10,480.00–556,993.00)	0.783	58,212.00 (11,992.00–689,662.00)	57,601.00 (6742.00–759,738.00)	0.323
TCBI [median(range)]	2259.00 (575.10–9463.00)	2285.00 (612.60–10,726.00)	0.539	1989.00 (575.10–16,780.00)	2111.00 (386.50–15,834.00)	0.628

HOMA-IR—Homeostasis Model Assessment of Insulin Resistance; HOMA%B—Homeostasis Model Assessment of β-cell Function; METS-IR—Metabolic Score for Insulin Resistance; TyG—Triglyceride-Glucose Index; CRI I—Castelli Risk Index I; CRI II—Castelli Risk Index II; LAP—Lipid Accumulation Product; LCI—Lipoprotein Combine Index; TCBI—triglyceride-total cholesterol-body weight index; * *p* < 0.05: statistically significant.

**Table 5 life-15-01845-t005:** Multiple linear regression of PreDM cohort.

Model 1						Model 2					
**Analysis** **of Variance**	**SS**	**DF**	**MS**	**F (DFn, DFd)**	***p*** **Value**	**Analysis** **of Variance**	**SS**	**DF**	**MS**	**F (DFn, DFd)**	***p*** **Value**
Regression	1586	3	528.7	F (3, 76) = 6.108	0.0009	Regression	2563	3	854.3	F (3, 76) = 11.59	<0.0001
HOMA-IR-PreDM	0.188	1	0.188	F (1, 76) = 0.002	0.962	HOMA%B-PreDM	19.09	1	19.09	F (1, 76) = 0.259	0.612
TyG-PreDM	1221	1	1221	F (1, 76) = 14.10	0.0003	LAP-PreDM	1338	1	1338	F (1, 76) = 18.15	<0.0001
BRI-PreDM	605	1	605	F (1, 76) = 6.990	0.010	WWI-PreDM	1698	1	1698	F (1, 76) = 23.04	<0.0001
**Parameter** **estimates**	**Variable**	**Estimate**	**95% CI (profile** **likelihood)**	**|t|**	***p*** **value**	**Parameter** **estimates**	**Variable**	**Estimate**	**95% CI (profile likelihood)**	**|t|**	***p*** **value**
β0	Intercept	−30.54	−66.69 to 5.620	1.682	0.096	β0	Intercept	66.17	53.99 to 78.35	10.82	<0.0001
β1	HOMA-IR-PreDM	−0.028	−1.249 to 1.192	0.046	0.962	β1	HOMA%B-PreDM	0.002	−0.005 to 0.009	0.509	0.612
β2	TyG-PreDM	7.480	3.513 to 11.45	3.756	0.0003	β2	LAP-PreDM	0.074	0.039 to 0.109	4.260	<0.0001
β3	BRI-PreDM	1.288	0.317 to 2.258	2.644	0.010	β3	WWI-PreDM	−2.501	−3.539 to −1.463	4.800	<0.0001

SS: Sum of Squares; DF: Degrees of Freedom; MS: Mean Square; DFn: numerator DF; DFd: denominator DF; CI: Confidence Interval.

**Table 6 life-15-01845-t006:** Multiple linear regression of T2DM cohort.

Model 1						Model 2					
**Analysis** **of Variance**	**SS**	**DF**	**MS**	**F (DFn, DFd)**	***p*** **Value**	**Analysis** **of Variance**	**SS**	**DF**	**MS**	**F (DFn, DFd)**	***p*** **Value**
Regression	10544	3	3515	F (3, 116) = 85.59	<0.0001	Regression	7620	3	2540	F (3, 116) = 38.33	<0.0001
HOMA-IR-T2DM	23.14	1	23.14	F (1, 116) = 0.563	0.454	HOMA%B-T2DM	98.78	1	98.78	F (1, 116) = 1.491	0.224
TyG-T2DM	1389	1	1389	F (1, 116) = 33.82	<0.0001	LAP-T2DM	4551	1	4551	F (1, 116) = 68.68	<0.0001
BRI-T2DM	6456	1	6456	F (1, 116) = 157.2	<0.0001	WWI-T2DM	143.5	1	143.5	F (1, 116) = 2.166	0.143
**Parameter** **estimates**	**Variable**	**Estimate**	**95% CI (profile** **likelihood)**	**|t|**	***p*** **value**	**Parameter** **estimates**	**Variable**	**Estimate**	**95% CI (profile** **likelihood)**	**|t|**	***p*** **value**
β0	Intercept	−23.41	−41.66 to −5.160	2.541	0.012	β0	Intercept	31.79	17.70 to 45.89	4.467	<0.0001
β1	HOMA-IR-T2DM	−0.072	−0.263 to 0.118	0.750	0.454	β1	HOMA%B-T2DM	−0.018	−0.048 to 0.011	1.221	0.224
β2	TyG-T2DM	5.947	3.922 to 7.973	5.815	<0.0001	β2	LAP-T2DM	0.112	0.085 to 0.139	8.287	<0.0001
β3	BRI-T2DM	3.316	2.792 to 3.839	12.54	<0.0001	β3	WWI-T2DM	1.011	−0.349 to 2.371	1.472	0.143

## Data Availability

The data used to support the findings of this study are available from the corresponding author upon reasonable request.

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
