# Peer review of "Beyond HOMA-IR: Comparative Evaluation of Insulin Resistance and Anthropometric Indices Across Prediabetes and Type 2 Diabetes Mellitus in Metabolic Syndrome Patients"

_life, 2025, doi:10.3390/life15121845_

Round 1

Reviewer 1 Report

Comments and Suggestions for Authors

Dear Editor,

Thank you for inviting me to review the manuscript entitled “Beyond HOMA-IR: Comparative Evaluation of Insulin Resistance and Anthropometric Indices Across Prediabetes and Type 2 Diabetes Mellitus in Metabolic Syndrome Patients” was revised.

There is evidence that anthropometric measurements can be associated with the risk of T2DM and metabolic syndrome development.  To date, no comprehensive consensus has been reached on which anthropometric index best evaluates the risk and predicts T2DM, particularly in older adults. Moreover, the combination with indexes of insulin resistance could be helpful for early diagnosis and for distinguishing these diseases.

The aim of the presented study is to evaluate the performance of non-insulin-based indices and advanced anthropometric measures in differentiating prediabetes from T2DM and their relation to β-cell function and cardiometabolic risk.

Overall, I find that the topic is interesting and relevant. The manuscript is well written and presents a wealthy set of results, related to the objective of the study.

I would like to point out the following comments and suggestions which could be considered by the authors in order to improve the manuscript.

Abstract

Please, write all terms in full when first mentioned before using abbreviations (e.g., HOMA-IR, METS-IR, TyG, BMI, BRI, and WWI – Lines 31-36, 67 – IR; Table 4 – liver enzymes: AST, ALT, ESR, PLT, Hb, CRP; Line 110 - HDL-c, Line 138- HbA1c, Line 306- HOMA%B) and provide consistency in the abbreviation used throughout the manuscript.

Introduction

The statement is not completely accurate. The determination of glucose and triglycerides (required for index calculation) are routine laboratory tests with well-defined pre-analytical requirements. Note that glucose and triglycerides are included in the calculation of METS-IR.

Lines 64- Please, use the abbreviations already introduced throughout the text.

Lines 111-112 Please, replace the word” superior” with a more appropriate term such as “more effective,” “better suited,” or “more accurate.”

Materials and Methods

Lines 122-123 My preposition is for these sentences to be removed.

Lines 125-127 The presented information is not very clear and inconsistent with Lines 133–135 and Abstract Lines 28–30. Please, check and correct.

Line 136 - Please, specify the upper age limit of participants.

Lines 138-139 - Please, indicate whether data were collected on corticosteroid or glucose-lowering drug use.

Lines 151-153 - Please, elucidate your approach to patients on lipid-lowering therapy. Were they included? If they are included, please, provide clarifications regarding the levels of the lipid profile elements. Total cholesterol and LDL-C are included in the calculation of Lipoprotein Combine Index – LCI Line 194 and Triglyceride-total cholesterol-body weight index- TCBI (TC only) - line 198.

Tables 1,2 and 3 should be removed, referencing is sufficient.

Lines 165-168 The information is repeated similar to that presented in lines 125-128, removing redundancy.

In this regard, I suggest to the authors of the study to prepare flow chart of subject participation

Lines 193-198, 205 Please, present the formulas along with the measurement units of the included indicators.

Laboratory-based assays

Line 208 Consider replacing 'biological sampling' with 'venous blood sampling'.

Lines 223-224 The flow cytometry principle is completely different from the Coulter method. Please, revise the sentence or remove it. Please, specify the types of tubes used for hematological analyses.

Lines 222-226 Consider rephrasing as “A complete blood count (CBC) and differential white blood cell count were performed using hematology analyzer "MINDRAY BC-6800”.

In this section are not presented lab analyzes, shown in the table 4 -AST, ALT, creatinine, CRP, ESR, HbA1c.  Please, express the  liver enzyme results in SI units (IU/L).  They  are in accordance with recent kinetic methods for their measurement.

The section Materials and Methods should be changed and rewritten to be more concise and clearer for understanding.

Results

Table 4 - “Hepatosteatosis” - Please, explain the diagnostic approach for this diagnosis - laboratory, instrumental, or combined.

“Hyperuricemia” - The term “hyperuricemia” is mentioned, but uric acid levels are not shown. Please, provide these values.

The results should be explained briefly and concisely, indicating only statistically significant differences throughout the section "Results"

Lines 294-299; 340-349; 413-422; 518-520. The pathophysiological interpretations for these variations belong in the Discussion section

Tables 5 and 6 displayed the results for the index” ABSI Z score” that is missing, along with the formula for calculation in the part "Assessment of Anthropometric and Adiposity Indices". Please, add this information.

Table 7: The TyG/HDL-c index is introduced but not described in the “Assessment of Lipemic Spectrum Indices.” Please, provide its definition and formula.

Lines 454, 469 - Consider replacing “notably” with “obviously”, or another suitable alternative.

There is significant room for improvement and rewriting the section “Results”.

Discussion

Lines 574-575 – The information is not accurate. The formula for METS-IR index calculation does not include waist circumference. Please, check and correct.

Lines 581-587 I propose that these sentences shall be removed. They present important scientific information but unrelated to the study’s purpose, methods, or findings.

Lines 629-631 Please, note that both indexes METS-IR and TyG were validated against the euglycemic hyperinsulinemic clamp. The reviewer proposes that it is unnecessary to perform this validation in every research study. Please, adjust wording.

Lines 631-633 - These sentences could be removed.

I suggest that the Discussions section can be rewritten.

Conclusions – could be shortened and refined to make it clearer, more focused, and directly reflective of the study findings.

Author Response

Thank you very much for all of your time and effort you took in reviewing our manuscript because it helped us improve our work. If there are any further remarks to be addressed please mention.

Abstract

Comment 1: Please, write all terms in full when first mentioned before using abbreviations (e.g., HOMA-IR, METS-IR, TyG, BMI, BRI, and WWI – Lines 31-36, 67 – IR; Table 4 – liver enzymes: AST, ALT, ESR, PLT, Hb, CRP; Line 110 - HDL-c, Line 138- HbA1c, Line 306- HOMA%B) and provide consistency in the abbreviation used throughout the manuscript.

Response 1: Revised. Also, the abstract was revised after the new modifications.

Introduction

Comment 2: The statement is not completely accurate. The determination of glucose and triglycerides (required for index calculation) are routine laboratory tests with well-defined pre-analytical requirements. Note that glucose and triglycerides are included in the calculation of METS-IR.

Response 2: Revised.

Comment 3: Lines 64- Please, use the abbreviations already introduced throughout the text.

Response 3: Revised, thank you!

Comment 4: Lines 111-112 Please, replace the word” superior” with a more appropriate term such as “more effective,” “better suited,” or “more accurate.”

Response 4: The word was replaced.

Materials and Methods

Comment 5: Lines 122-123 My preposition is for these sentences to be removed.

Response 5: We removed the sentences.

Comment 6: Lines 125-127 The presented information is not very clear and inconsistent with Lines 133–135 and Abstract Lines 28–30. Please, check and correct.

Response 6: We clarified and added the flowchart.

Comment 7: Line 136 - Please, specify the upper age limit of participants.

Response 7: We now mention the upper age limit.

Comment 8: Lines 138-139 - Please, indicate whether data were collected on corticosteroid or glucose-lowering drug use.

Response 8: Mentioned.

Comment 9: Lines 151-153 - Please, elucidate your approach to patients on lipid-lowering therapy. Were they included? If they are included, please, provide clarifications regarding the levels of the lipid profile elements. Total cholesterol and LDL-C are included in the calculation of Lipoprotein Combine Index – LCI Line 194 and Triglyceride-total cholesterol-body weight index- TCBI (TC only) - line 198.

Response 9: We mentioned the patients were undergoing lipid lowering therapy.

Comment 10: Tables 1,2 and 3 should be removed, referencing is sufficient.

Response 10: We would like not to remove these tables due to the fact that they provide an easier way to read and see the exact criteria.

Comment 11: Lines 165-168 The information is repeated similar to that presented in lines 125-128, removing redundancy.

In this regard, I suggest to the authors of the study to prepare flow chart of subject participation

Response 11: We now added the flowchart, please check figure 1.

Comment 12: Lines 193-198, 205 Please, present the formulas along with the measurement units of the included indicators.

Response 12: Added.

Laboratory-based assays

Comment 13: Line 208 Consider replacing 'biological sampling' with 'venous blood sampling'.

Response 13: Replaced.

Comment 14: Lines 223-224 The flow cytometry principle is completely different from the Coulter method. Please, revise the sentence or remove it. Please, specify the types of tubes used for hematological analyses.

Response 14: We rephrased.

Comment 15: Lines 222-226 Consider rephrasing as “A complete blood count (CBC) and differential white blood cell count were performed using hematology analyzer "MINDRAY BC-6800”.

Response 15: Thank you for the suggestions, we used your rephrasing.

Comment 16: In this section are not presented lab analyzes, shown in the table 4 -AST, ALT, creatinine, CRP, ESR, HbA1c.  Please, express the  liver enzyme results in SI units (IU/L).  They  are in accordance with recent kinetic methods for their measurement.

Comment 16: Added, thank you for the observation.

Comment 17: The section Materials and Methods should be changed and rewritten to be more concise and clearer for understanding.

Response 17: The section was revised.

Results

Comment 18: Table 4 - “Hepatosteatosis” - Please, explain the diagnostic approach for this diagnosis - laboratory, instrumental, or combined.

Response 18: The mention of hepatosteatosis and hyperuricemia is based on the patient's personal medical history. They were already diagnosed before our study; we did not diagnose them now. Thank you for mentioning it.

Comment 19: “Hyperuricemia” - The term “hyperuricemia” is mentioned, but uric acid levels are not shown. Please, provide these values.

Response 19: The mention of hepatosteatosis and hyperuricemia is based on the patient's personal medical history. They were already diagnosed before our study; we did not diagnose them now. Thank you for mentioning it.

Comment 20: The results should be explained briefly and concisely, indicating only statistically significant differences throughout the section "Results"

Response 20: We shortened the result section. Thank you for the suggestion!

Comment 21: Lines 294-299; 340-349; 413-422; 518-520. The pathophysiological interpretations for these variations belong in the Discussion section

Response 21: We moved them to the discussion section. Thank you for the idea!

Comment 22: Tables 5 and 6 displayed the results for the index” ABSI Z score” that is missing, along with the formula for calculation in the part "Assessment of Anthropometric and Adiposity Indices". Please, add this information.

Response 22: ABSI Z score is the last one in the table, under ABSI; we mentioned in the methods what we used for it.

Comment 23: Table 7: The TyG/HDL-c index is introduced but not described in the “Assessment of Lipemic Spectrum Indices.” Please, provide its definition and formula.

Response 23: We did not introduced any definition and formula because it is obvious that is the ratio between TyG (for which we provided a formula) and HDL-c (which is a value). If you consider it necessary we can add it. Thank you!

Comment 24: Lines 454, 469 - Consider replacing “notably” with “obviously”, or another suitable alternative.

Response 24: Replaced.

There is significant room for improvement and rewriting the section “Results”.

Discussion

Comment 25: Lines 574-575 – The information is not accurate. The formula for METS-IR index calculation does not include waist circumference. Please, check and correct.

Response 25: Thank you very much for the observation, it was such a help. We excluded the waist from the sencence.

Comment 26: Lines 581-587 I propose that these sentences shall be removed. They present important scientific information but unrelated to the study’s purpose, methods, or findings.

Response 26: Removed.

Comment 27: Lines 629-631 Please, note that both indexes METS-IR and TyG were validated against the euglycemic hyperinsulinemic clamp. The reviewer proposes that it is unnecessary to perform this validation in every research study. Please, adjust wording.

Response 27: Removed.

Comment 28: Lines 631-633 - These sentences could be removed.

Response 28: Removed, thank you!

Comment 29: I suggest that the Discussions section can be rewritten.

Response 29: Revised. Please let us know if it still needs further rewriting.

Comment 30: Conclusions – could be shortened and refined to make it clearer, more focused, and directly reflective of the study findings.

Response 30: We refined the conclusions section completely. Thank you!

Reviewer 2 Report

Comments and Suggestions for Authors

Managing the continuing rise in type 2 diabetes, and associated pre-diabetes, is a challenge for national health strategies and the limited resources.  HOMA-IR tests are widely used to confirm a diagnosis of diabetes, but involve measurement of insulin. METS-IR scoring, based on measurement of four metabolic parameters: FPG, TG, BMI, and HDL, was recently introduced as an alternative.

The article presents a detailed comparison between METS-IR, and other indices, with more established methods of diagnosing diabetes.  It includes detailed analysis of data from two cohorts of patients, diabetic and pre-diabetic, but does not include a control group.

Criteria for diagnosis of pre-diabetes or type 2 diabetes are clearly described, as are criteria for inclusion.  A large number of clinical and metabolic parameters are measured across the cohorts. The data are analysed extensively.

Some striking differences between cohorts were recorded in Table 4. The description of the data is presented in a series of key points, section by section, which is succinct and effective.  Similarly for other tables.

Some points to address:

Tables 5 and 6 are based on grouping data according to normal/altered HOMA-IR and METS-IR, giving four groupings in total.  The basis for these groupings needs to be explained - they appear to represent a progression of disease, but it's not clear why they should so. Please clarify what is meant by "normal" and "altered" and explain the basis for the sub-grouping.

There appears to be no direct comparison of HOMA-IR and METS-IR between prediabetic and diabetic groups.  The authors should consider linear regression analysis between HOMA-IR and METS-IR.  This links to the discussion lines 526-7; it is not clear what data are being discussed.  

References and citations can be improved: citations are grouped together too much e.g. 1-6, 7-11 etc.  Normally expect citation next to item described; the first description of METS-IR is ref [23] - why is not included in the introduction.  Ref [24] is not peer reviewed.

Author Response

Thank you for your thorough review of our manuscript, which significantly contributed to enhancing the quality of our work. Please let us know if there are any additional comments or concerns.

Comment 1: Tables 5 and 6 are based on grouping data according to normal/altered HOMA-IR and METS-IR, giving four groupings in total.  The basis for these groupings needs to be explained - they appear to represent a progression of disease, but it's not clear why they should so. Please clarify what is meant by "normal" and "altered" and explain the basis for the sub-grouping.

Response 1: Revised.

Comment 2: There appears to be no direct comparison of HOMA-IR and METS-IR between prediabetic and diabetic groups.  The authors should consider linear regression analysis between HOMA-IR and METS-IR.  This links to the discussion lines 526-7; it is not clear what data are being discussed.  

Reponse 2: We added the regression.

Comment 3: References and citations can be improved: citations are grouped together too much e.g. 1-6, 7-11 etc.  Normally expect citation next to item described; the first description of METS-IR is ref [23] - why is not included in the introduction.  Ref [24] is not peer reviewed.

Response 3: Revised.

Round 2

Reviewer 1 Report

Comments and Suggestions for Authors

x

Dear Editor,

The revised manuscript with the “Beyond HOMA-IR: Comparative Evaluation of Insulin Re sistance and Anthropometric Indices Across Prediabetes and Type 2 Diabetes Mellitus in Metabolic Syndrome Patients” was checked.

I would like to express my gratitude to the authors for the corrections they have made on the basis of the initial feedback. I am satisfied with the author's responses to my comments/questions raised in my initial review. The authors have considered the suggestions appropriately and have done a good job of almost all points, responding to the reviewer's comments and incorporating them into their revised manuscript.

The corrections were made in all sections of the manuscript, including Abstract (it was rewritten), Introduction, Materials and Methods, Statistical analysis, Results (additional approaches and tables 8 and 9), Discussion and Conclusions. The specified sections which have been corrected or additional information has been added improved the comprehensibility of the manuscript and enhanced its quality.

I would like to highlight several comments and suggestions regarding the revised manuscript:

  1. When you are writing all terms in full at first mentioned and after that abbreviating, please, use the abbreviations and provide consistency in their usage throughout the manuscript (e.g., T2DM – line 51; HOMA-IR-line 53; HOMA-B-line 284; TyG-line 54; METS-IR-line 58; BRI-line 58; WWI-line 59; PreDM-line 64, BMI-line 82). All of these abbreviations are introduced in the abstract.
  2. This is a research paper focusing on a clinical model for comparative evaluation of IR and anthropometric indices across prediabetes and T2DM in patients with MetS. ADA recommendations for diagnosis of T2DM and pre-diabetes (and the IDF recommendations for MetS) are generally accepted and need not be presented in a scientific publication. From this perspective, I firmly recommend to the authors of the manuscript to remove Tables 1, 2 and 3. The references are sufficient.
  3. Please, include the ratio TyG/HDL-c in the section 2.4. "Assessment of Lipemic Spectrum Indices" along with the appropriate reference. If this ratio represents the authors’ own view, please, justify its inclusion with appropriate explanations.

  1. Regarding patients with hepatosteatosis and hyperuricemia. I assume that they have laboratory analyses in support of these diagnoses in their personal medical    Please, confirm this information.  If such evidence is not available, I suggest removing the variables from Table 4 (“General Characteristics of the Study Cohorts”).
  2. Please, specify the measurement units for insulin (Table 4) and for HbA1c ( Tables 5 and 6).

Author Response

Thank you very much for your help and time!

Comment 1: When you are writing all terms in full at first mentioned and after that abbreviating, please, use the abbreviations and provide consistency in their usage throughout the manuscript (e.g., T2DM – line 51; HOMA-IR-line 53; HOMA-B-line 284; TyG-line 54; METS-IR-line 58; BRI-line 58; WWI-line 59; PreDM-line 64, BMI-line 82). All of these abbreviations are introduced in the abstract.

Response 1: We followed the MDPI instructions: "Acronyms/Abbreviations/Initialisms should be defined the first time they appear in each of three sections: the abstract; the main text; the first figure or table. When defined for the first time, the acronym/abbreviation/initialism should be added in parentheses after the written-out form." We understand from your comment that you want to delete the explanation from the lines mentioned because they were already mentioned in Abstract;  however it is contrary to MDPI instructions. If we misunderstood your comment please mention.

Comment 2: This is a research paper focusing on a clinical model for comparative evaluation of IR and anthropometric indices across prediabetes and T2DM in patients with MetS. ADA recommendations for diagnosis of T2DM and pre-diabetes (and the IDF recommendations for MetS) are generally accepted and need not be presented in a scientific publication. From this perspective, I firmly recommend to the authors of the manuscript to remove Tables 1, 2 and 3. The references are sufficient.

Response 2: Revised after your suggestions.

Comment 3: Please, include the ratio TyG/HDL-c in the section 2.4. "Assessment of Lipemic Spectrum Indices" along with the appropriate reference. If this ratio represents the authors’ own view, please, justify its inclusion with appropriate explanations.

Response 3: Revised.

Comment 4: Regarding patients with hepatosteatosis and hyperuricemia. I assume that they have laboratory analyses in support of these diagnoses in their personal medical    Please, confirm this information.  If such evidence is not available, I suggest removing the variables from Table 4 (“General Characteristics of the Study Cohorts”).

Response 4: Removed.

Comment 5: Please, specify the measurement units for insulin (Table 4) and for HbA1c ( Tables 5 and 6).

Response 5: Revised.